# Generative Retrieval Meets Multi-Graded Relevance

**Yubao Tang**[1,2]   **Ruqing Zhang**[1,2]   **Jiafeng Guo**[1,2*]   **Maarten de Rijke**[3]
**Wei Chen**[1,2]   **Xueqi Cheng**[1,2]

[1]CAS Key Lab of Network Data Science and Technology, ICT, CAS
[2]University of Chinese Academy of Sciences
[3]University of Amsterdam
`{tangyubao21b,zhangruqing,guojiafeng,chenwei2022,cxq}@ict.ac.cn`
`m.derijke@uva.nl`

## Abstract

Generative retrieval represents a novel approach to information retrieval. It uses an encoder-decoder architecture to directly produce relevant document identifiers (docids) for queries. While this method offers benefits, current approaches are limited to scenarios with binary relevance data, overlooking the potential for documents to have multi-graded relevance. Extending generative retrieval to accommodate multi-graded relevance poses challenges, including the need to reconcile likelihood probabilities for docid pairs and the possibility of multiple relevant documents sharing the same identifier. To address these challenges, we introduce a framework called GRaded Generative Retrieval ($GR^2$). $GR^2$ focuses on two key components: ensuring relevant and distinct identifiers, and implementing multi-graded constrained contrastive training. First, we create identifiers that are both semantically relevant and sufficiently distinct to represent individual documents effectively. This is achieved by jointly optimizing the relevance and distinctness of docids through a combination of docid generation and autoencoder models. Second, we incorporate information about the relationship between relevance grades to guide the training process. We use a constrained contrastive training strategy to bring the representations of queries and the identifiers of their relevant documents closer together, based on their respective relevance grades. Extensive experiments on datasets with both multi-graded and binary relevance demonstrate the effectiveness of $GR^2$.

## 1  Introduction

Generative retrieval (GR) [51] is a new paradigm for information retrieval (IR), where all information in a corpus is encoded into the model parameters and a ranked list is directly produced based on a single parametric model. In essence, a sequence-to-sequence (Seq2Seq) encoder-decoder architecture is used to directly predict identifiers (docids) of documents that are relevant to a given query. Recent studies have achieved impressive retrieval performance on many search tasks [15, 53, 67, 92].

Current work on GR mainly focuses on binary relevance scenarios, where a binary division into relevant and irrelevant categories is assumed [20, 52, 83], and a query is usually labeled with a single relevant document [38, 57] or multiple relevant documents that have the same relevance grade [33]. The standard Seq2Seq objective, via maximizing likelihood estimation (MLE) of the output sequence with teacher forcing, has been used extensively in GR due to its simplicity. However, in real-world search scenarios, documents may have different degrees of relevance [18, 71, 72, 82, 91] as binary relevance may not be sufficiently represent fine-grained relevance. In traditional learning-to-rank (LTR), multi-graded relevance judgments [23, 69] are widely considered, with nDCG [35] and ERR [12] being particularly popular. In modeling multi-graded relevance in LTR, a popular approach is

---

*Corresponding author.

the pairwise method [10, 84] which involves weighting different documents based on their relevance grades and predicting the relative order of a document pair.

Compared to common LTR algorithms, the learning objective currently being used in GR differs significantly: the standard Seq2Seq objective emphasizes one-to-one associations between queries and docids, aiming to generate a single most relevant docid. Inspired by pairwise methods in LTR, a straightforward approach to extending GR to multiple grades, involves having the GR model generate the likelihood of docids with higher relevance grades being greater than that of lower relevance grades. The docid likelihood is the product of the likelihoods of each token in the generated docid. Docids commonly exhibit distinct lengths, as a fixed length might not adequately encompass diverse document semantics. However, the variation in docid lengths within the corpus may lead to smaller likelihood scores for longer docids. Although some GR work [44, 78, 92, 93, 102] use a pairwise or listwise loss for optimization, they still only consider binary relevance or require complex multi-stage optimization. Besides, essential topics in multi-graded relevant documents may be similar, emphasizing the need for a one-to-one correspondence between document content and its identifier to ensure distinctness. Consequently, harnessing a GR model's capabilities for multi-graded relevance ranking in a relatively succinct manner remains an non-trivial challenge.

To this end, we consider *multi-graded generative retrieval* and propose a novel GRaded Generative Retrieval ($GR^2$) framework, with three key features:

(1) To enhance docid distinctness while ensuring its relevance to document semantics, we introduce a *regularized fusion* approach with two modules: (i) a *docid generation module*, that produces pseudo-queries based on the original documents as the docids; and (ii) an *autoencoder module*, that reconstructs the target docids from their corresponding representations. We train them jointly to ensure that the docid representation is close to its corresponding document representation while far from other docid representations.

(2) For the mapping from a query to its relevant docids, we design a *multi-graded constrained contrastive* (MGCC) loss to capture the relationships between labels with different relevance grades. Considering the incomparability of likelihood probabilities associated with docids of varying lengths, we convert queries and docids into representations within the embedding space. The core idea is to pull the representation of a given query in the embedding space towards those of its relevant docids, while simultaneously pushing it away from representations of irrelevant docids in the mini-batch. To maintain the order between relevance grades in the embedding space, the strength of the pull is determined by the relevance grades of the docids. The distinction between MGCC and pairwise methods in LTR [9, 10, 84] lies in proposing more specific grade penalties and constraints to regulate the relative distances between query representations and docid representations of different grades.

(3) We explore two learning scenarios, i.e., *supervised learning* and *pre-training*, to learn generative retrieval models using our $GR^2$ framework. Importantly, our method for obtaining docids is applicable to both multi-graded and binary relevance data, and it can reduce to the supervised contrastive approach in binary relevance scenarios.

Our main contributions are: (i) We introduce a general $GR^2$ framework for both binary and multi–graded relevance scenarios, by designing relevant and distinct docids and using the information about the relationship between labels. (ii) Through experiments on 5 representative document retrieval datasets, $GR^2$ achieves 14% relative significant improvements for P@20 on Gov 500K dataset over the SOTA GR baseline RIPOR [85]. (iii) Even in low-resource scenarios, our method performs well, surpassing BM25 on two datasets. On large-scale datasets, it achieves comparable results to RIPOR.

## 2   Related Work

**Learning to rank (LTR).** LTR ranks candidate documents using ranking functions for queries, employing pointwise, pairwise, and listwise approaches. In the pointwise approach, a query is modeled with a single document, similar to using MLE in GR, making it challenging to capture global associations. Pairwise LTR treats document pairs as instances, e.g., LambdaRank [10], LTRGR [44], and RIPOR [92]. The MGCC loss proposed here aligns with a pairwise approach. The listwise method [78, 88] treats entire document lists as instances, involving high optimization costs.

**Generative retrieval.** GR has been proposed as a new paradigm for IR in which documents are returned using model parameters only [51]. A single model can directly generate relevant documents

for a query. Inspired by this blueprint, there have been several proposals [7, 14, 22, 40, 43, 79, 85] to learn a Seq2Seq model by simultaneously addressing the two key issues below.

*Key issue 1: Building associations between documents and docids.* The widely-used docid designs are pre-defined and learnable docids [77]. Pre-defined docids are fixed during training, e.g., document titles [22], semantically structured strings [56, 79, 85], n-grams [7, 14, 44], pseudo-queries [76], URLs [68, 102, 104], product quantization code [13, 104]. Learnable docids are tailored to retrieval tasks, e.g., discrete numbers [74], important word sets [86, 96, 97] and residual quantization code [92, 93]. Though effective in some tasks, these docid designs have limitations. Titles and URLs rely on metadata, while n-grams demand storage of all n-grams. Quantization codes lack interpretability. Learning optimal learnable docids is challenging, involving a complex learning process. Considering performance, implementation complexity, and storage requirements, the pre-defined pseudo-query is a promising compromise choice. Unfortunately, semantically similar documents might have similar docids or even repetitions, making it challenging for the GR model to distinguish them in the both binary and multi-graded relevance scenarios. To generate diverse and relevant docids, we propose a docid fusion approach based on pseudo-queries.

*Key issue 2: Mapping queries to relevant docids.* Given a query, a GR model takes as input a query and outputs its relevant docids by maximizing the output sequence likelihood, which is only suitable for binary relevance. If a query has only one relevant document, it is paired with its relevant docid. If a query has multiple relevant documents at the same grade, it is paired with multiple relevant docids. The relative order of relevant docids in the returned list is random. Such a learning objective cannot handle search tasks with multi-graded relevance, which limits its efficacy for general IR problems. While certain GR studies [44, 92, 93, 102] employ a pairwise or listwise loss for optimization, they remain limited to binary relevance or necessitate intricate multi-stage optimization processes. In this work, we use all available relevance labels to enable multi-graded GR. For more related work, please refer to Appendix C.

## 3 Preliminaries

**Document retrieval.** Assume that $L_q = [1, \ldots, l, \ldots, L]$ is the grade set representing different degrees of relevance. We assume that there exists a total order between the grades $l > l-1 > \cdots > 1$, $\forall l \in L_q$. Let $q$ be a query from the query set $Q$, and $D_q = \{d_1, d_2, \ldots, d_N\}$ be the set of $N$ relevant documents for $q$, which are selected from the large document collection $D$. $D_Q$ is the relevant document set for $Q$. We write $D_q^l$ for the documents with grade $l$ for $q$. The document retrieval task is to find a retrieval model $f$ to produce the ranked list of relevant documents for the given query, i.e., $\pi_f(q) := [\arg\max_d^{(1)} f(q, d), \arg\max_d^{(2)} f(q, d), \ldots]$, where $\arg\max_d^{(i)} f(q, d)$ denotes the $i$-ranked document $d$ for $q$ over $D$ given by $f$ via matching the query and documents. The model $f$ is optimized by minimizing its loss function over some labeled datasets, i.e., $\min_f \sum_Q \sum_D \mathcal{L}(f; q, D_q)$.

**Multi-graded generative retrieval.** In an end-to-end architecture, the GR process directly returns a ranked list for a given query, without a physical index component. Assume that the indexing mechanism to represent docids is $I : D \to I_D$, where $I_D$ is the corresponding docid set. For the observed relevant document set $D_q, q \in Q$, the indexing mechanism $I$ maps them to the docid set $I_{D_q} = \{I_{D_q^l} \mid l = 1, \ldots, L\}$, in which the docid $id^l \in I_{D_q^l}$ is at grade $l$. The GR model observes pairs of a query and docid with multi-graded relevance labels under the indexing mechanism $I$, i.e., $\{Q, I_{D_Q}\}$, where $I_{D_Q}$ denotes $\{I_{D_q^l} \mid l = 1, \ldots, L, q \in Q\}$. Given $Q$, the model $g : Q \to I_{D_Q}$ autoregressively generates a ranked list of candidate docids in descending order of output likelihood conditioned on each query. Mathematically, $g(q; \theta) = P_\theta(id \mid q) = \prod_{t \in [1, |id|]} p_\theta(w_t \mid q, w_{<t})$, where $w_t$ is the $t$-th token in the docid $id \in I_D$ and $w_{<t}$ represents all tokens before the $t$-th token in $id$. $\theta$ is the model parameters. During inference, the GR model produces the ranked docid list via $\pi_g(q) := [g^{(1)}(q), g^{(2)}(q), \ldots] = [\arg\max_{id}^{(1)} P_\theta(id \mid q), \arg\max_{id}^{(2)} P_\theta(id \mid q), \ldots]$, where $\arg\max_{id}^{(i)} P_\theta(id \mid q)$ denotes $id$ for $q$ whose generation likelihood is ranked at position $i$.

## 4 Methodology

In this section, we develop a general GR framework called GR$^2$ to support multi-graded relevance learning. We address two main challenges: (i) how to generate relevant and distinct identifiers given the original documents (Section 4.1), and (ii) how to capture the interrelationship between docids in a ranking for a query (Section 4.2). Next, we introduce the optimization process (Section 4.3).

## 4.1 Docid design: regularized fusion approach

A popular docid representation method is to employ a query generation (QG) technique [60] to generate a pseudo-query conditioned on the document as the docid [43, 76]. It is common for different documents to share identical or similar docids when they contain similar information [7, 14, 76]. While this similarity can aid the GR model in recognizing the likeness, it also poses a challenge when the GR model needs to differentiate among multiple documents with varying relevance grades to a query. Therefore, as shown in Figure 5 in Appendix A, we propose a regularized fusion approach to optimize the trade-off between relevance and distinctness in docids.

The key idea is to jointly optimize the relevance and distinctness that fuses the latent space of a QG model, i.e., a docid generation model, and that of an autoencoder (AE) model. The AE model is used to reconstruct the target query, and both models are based on an encoder-decoder architecture. We share the same decoder for both QG and AE models as in [29, 46]. Specifically, we propose two simple yet effective auxiliary regularization terms, i.e., a relevance term and a distinctness term.

**Relevance regularization term.** To improve the relevance [36, 90], we encourage the representation of a document and that of the corresponding docid (i.e., pseudo-query) to be close to each other in the shared latent space. We also aim to increase the distance between the representation of a document and that of irrelevant docids associated with other documents. This term is formalized as:

$$\mathcal{L}_{Rel}(Q, D_Q; \theta_{QG}, \theta_{AE}) = -\frac{1}{|Q|} \sum_{q \in Q, d \in D_Q} \frac{\exp(sim(e_{QG}^d, e_{AE}^q))}{\exp(sim(e_{QG}^d, e_{AE}^q)) + \zeta}, \tag{1}$$

where $\zeta = \sum_{d \in D_Q, \overline{q} \in Q, \overline{q} \neq q} \exp(sim(e_{QG}^d, e_{AE}^{\overline{q}}))$. For each query-document pair, $e_{QG}^d$ is the document representation obtained by the encoder of the QG model, and $e_{AE}^q$ is the query representation obtained by the encoder of the AE model; $\overline{q}$ is one of the remaining queries except for $q$ in the batch, and the batch size is $|Q|$; $sim(\cdot, \cdot)$ is the dot-product function; and $\theta_{QG}$ and $\theta_{AE}$ are model parameters of the QG model and AE model, respectively.

**Distinctness regularization term.** To enhance the distinctness between documents and between docids, we push away the representations of different documents in the document space and, simultaneously, push away the representations of different docids in the docid space. Additionally, to establish a connection between the two latent spaces of docids and documents, we ensure that the representation of a document and its corresponding docid are close in the same latent space. In a batch, the distinctness regularization term $\mathcal{L}_{Div}(Q, D_Q; \theta_{QG}, \theta_{AE})$ is formalized as:

$$\mathcal{L}_{Div}(\cdot) = \sum_{d, \overline{d} \in D_Q, d \neq \overline{d}} \frac{sim(e_{QG}^d, e_{QG}^{\overline{d}})}{|Q|(|Q| - 1)} + \sum_{q, \overline{q} \in Q, q \neq \overline{q}} \frac{sim(e_{AE}^q, e_{AE}^{\overline{q}})}{|Q|(|Q| - 1)} - \sum_{q \in Q, d \in D_Q} \frac{sim(e_{QG}^d, e_{AE}^q)}{|Q|}, \tag{2}$$

where $\overline{d}$ is an irrelevant document with respect to $q$ in the batch. We include a discussion on the difference between the two regularization terms in Appendix B.

**Jointly training the QG and AE model.** Both models use MLE to optimize their targets based on their inputs. Therefore, the overall optimization objective $\mathcal{L}_{Docid}(Q, D_Q; \theta_{QG}, \theta_{AE})$ is:

$$\mathcal{L}_{Docid}(\cdot) = \mathcal{L}_{MLE}^{QG}(Q, D_Q; \theta_{QG}) + \mathcal{L}_{MLE}^{AE}(Q; \theta_{AE}) + \alpha \mathcal{L}_{Rel}(\cdot) + \beta \mathcal{L}_{Div}(\cdot), \tag{3}$$

where $\mathcal{L}_{MLE}^{QG}(\cdot) = -\sum_{q \in Q, d \in D_Q} \log P_{\theta_{QG}}(q|d)$, and $\mathcal{L}_{MLE}^{AE}(\cdot) = -\sum_{q \in Q} \log P_{\theta_{AE}}(q|e_{AE}^q)$. $P_{\theta_{QG}}(q|d)$ and $P_{\theta_{AE}}(q|e_{AE}^q)$ denote the output query likelihood conditioned on the document, and $e_{AE}^q$. $\alpha$ and $\beta$ are hyperparameters.

**Relevant and distinct docid generation.** During inference, following [29, 100], we sample different latent vectors to generate docids. Given a document, we obtain its representation $e_{QG}^d$ by the QG model's encoder. We introduce a random vector $r$ that is uniformly sampled from a hypersphere of radius $|r|$ centered at $e_{QG}^d$, denoted as $z_{d,r} = e_{QG}^d + r$. The value of $|r|$ is tuned on the validation set to optimize the trade-off between relevance and distinctness. $z_{d,r}$ is then used as the initial state for the decoder of QG model. We subsequently generate a list of pseudo-queries using beam search decoding. Initially, we choose the top 1 pseudo-query as the docid. If there are still duplicate docids,

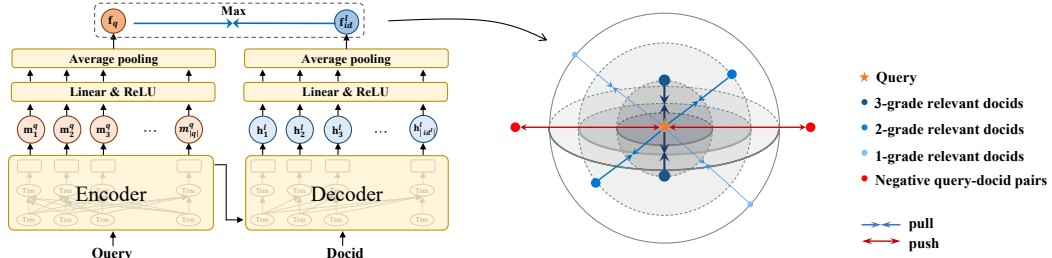

Figure 1: A Seq2Seq encoder-decoder architecture is used to consume queries and produce relevant docids for GR. We employ a multi-graded constrained contrastive loss (Section 4.2) to characterize the relationships among relevance labels based on the relevant and distinct docids (Section 4.1).

we select subsequent pseudo-queries from the list to replace these duplicates until all docids in the corpus are unique. According to our experimental analysis, selecting up to the top 2 can ensure docid uniqueness in the corpus. In this way, we first generate diverse docids $id$ relevant to the original text, serving as the basis for GR model learning. Considering learning costs, the models for docid design and the GR model are distinct. Though not end-to-end, These fixed docids can guide the GR model towards appropriate optimization, whereas joint optimization increases the learning difficulty.

## 4.2 Multi-graded constrained contrastive loss

After obtaining docids, we introduce the *multi-graded constrained contrastive* (MGCC) loss for the GR model. Positive pairs and negative pairs are constructed by pairing each query with its relevant docids drawn from all grades, and with all docids relevant to the rest of queries in the mini-batch except it, respectively. As illustrated in Figure 1, the key idea is to force positive pairs closer together in the representation space, but the magnitude of the force is dependent on the relevance grade. The MGCC loss includes a grade penalty and constraint.

**Grade penalty.** To distinguish between multiple positive pairs, our key idea is to apply higher penalties to positive pairs constructed from higher grades, forcing them closer than negative pairs constructed from lower grades. We first define the loss $\mathcal{L}_{Pair}(q, id^l; \theta)$ between a query $q$ and its relevant docid at grade $l$, as

$$\mathcal{L}_{Pair}(q, id^l; \theta) = \log \frac{\exp(sim(\mathbf{f}_q, \mathbf{f}_{id}^l)/\tau)}{\sum_{a \in A_q} \mathbb{1}_{[\mathbf{f}_q \neq \mathbf{f}_a]} \exp(sim(\mathbf{f}_q, \mathbf{f}_a)/\tau)}, \tag{4}$$

where $A_q$ includes all positive query-docid pairs at different grades and other negative query-docid pairs for $q$. $\mathbf{f}_q$ and $\mathbf{f}_{id}^l$ denote the representation of $q$ and $id^l$, respectively. They are computed based on the encoder and decoder hidden states, respectively, i.e.,

$$\mathbf{f}_q = \xi(\mathbf{M}^q; \theta), \quad \mathbf{f}_{id}^l = \xi(\mathbf{H}^l; \theta), \tag{5}$$

$$\xi([\mathbf{v}_1, \ldots, \mathbf{v}_T; \theta]) = \text{AvgPool}([\mathbf{u}_1, \ldots, \mathbf{u}_T]), \tag{6}$$

$$\mathbf{u}_t = \text{ReLU}(\mathbf{W}\mathbf{v}_t + \mathbf{b}), \tag{7}$$

where $\xi$ is the composition of affine transformation with the ReLU [54] and average pooling. $\mathbf{H}^l = [\mathbf{h}_1^l, \ldots, \mathbf{h}_{|id^l|}^l]$ is a concatenation of the decoder hidden states of $id^l$. $\mathbf{M}^q = [\mathbf{m}_1^q, \ldots, \mathbf{m}_{|q|}^q]$ is the concatenation of the hidden representations generated by the encoder of $q$. In this way, the loss for $Q$ is: $\sum_{q \in Q} \frac{1}{L} \sum_{l=1}^{L} \frac{-\lambda_l}{|I_{D_q^l}|} \sum_{id^l \in I_{D_q^l}} \mathcal{L}_{Pair}(q, id^l; \theta)$, where $id^l \in I_{D_q^l}$ is a docid at relevance grade $l$ for $q$; $\lambda_l$ is a controlling parameter that applies a fixed penalty for each grade, contributing to preserving the relevance level explicitly.

**Grade constraint.** Inspired by the hierarchical constraint in classification [30, 98], where a class higher in the hierarchy cannot have a lower confidence score than a class lower in the ancestry sequence, for each $q$, we propose to enforce a grade constraint $\mathcal{L}_{Max}$, i.e., the maximum loss from all positive pairs at grade $l$:

$$\mathcal{L}_{Max}(l, q, id^l) = \max_{(q, id^l; \theta)} \mathcal{L}_{Pair}(q, id^l; \theta). \tag{8}$$

The loss between query-docid pairs constructed from a higher relevance grade will never be higher than that constructed from a lower relevance grade. The final MGCC loss $\mathcal{L}_{MGCC}(Q, I_{D_Q}; \theta)$ is:

$$\mathcal{L}_{MGCC}(\cdot) = \sum_{q \in Q} \frac{1}{L} \sum_{l=1}^{L} \frac{-\lambda_l}{|I_{D_q^l}|} \sum_{id^l \in I_{D_q^l}} \max(\mathcal{L}_{Pair}(q, id^l; \theta), \mathcal{L}_{Max}(l+1, q, id^{l+1}; \theta)). \quad (9)$$

For binary relevance datasets, i.e., where there is only a single level of relevance labels (i.e., $L = 1$), the MGCC loss reduces to the supervised contrastive loss [16], which helps force the representation of the query close to that of its relevant documents, while far away from other irrelevant documents. In this way, $GR^2$ can also tackle the GR problem for the scenarios with binary relevance data, and this notably reduces complexity compared to multi-graded relevance.

### 4.3 Learning and optimization

**Supervised learning.** Based on the docids, we directly supervise the GR model with $\mathcal{L}_{MGCC}$, and we denote this version as $GR^{2S}$. To index all documents in a corpus, we adopt the MLE loss to learn document-docid pairs[79]. To guarantee the generation of each relevant docid to a query, we adopt the MLE for query-docid pairs at different grades. The final supervised learning loss is:

$$\mathcal{L}_{total}(Q, D, I_D; \theta) = \gamma \mathcal{L}_{MGCC}(Q, I_{D_Q}; \theta) + \mathcal{L}_{MLE}^q(Q, I_{D_Q}; \theta) + \mathcal{L}_{MLE}^d(D, I_D; \theta), \quad (10)$$

$$\mathcal{L}_{MLE}^q(Q, I_{D_Q}; \theta) = -\sum_{q \in Q} \frac{1}{L} \sum_{l=1}^{L} \frac{1}{|I_{D_q^l}|} \sum_{id^l \in I_{D_q^l}} \log P_\theta(id^l \mid q), \quad (11)$$

and $\mathcal{L}_{MLE}^d(D, I_D; \theta) = -\sum_{d \in D} \log P_\theta(id \mid d)$, where $\gamma$ is a hyperparameter, $P_\theta(id^l \mid q)$ and $P_\theta(id \mid d)$ denote the output docid likelihood conditioned on the query and document, respectively.

The learning objective currently being used in GR is usually defined as $\mathcal{L}_{MLE}^q(Q, I_{D_Q}; \theta) + \mathcal{L}_{MLE}^d(D, I_D; \theta)$, which does not capture the relationships between labels.

**Pre-training and fine-tuning.** We also explore the use of $GR^2$ in a pre-training scenario. To construct pre-training data, we use the English Wikipedia [87] to build a set of pseudo-pairs of queries and docids. We use the unique titles of Wikipedia articles as the docids for pre-training and assume that a random sentence in the abstract can be viewed as a representative query of the article.

Then, for each query, we construct its relevant documents with 4 relevance grades as follows, and leave other grades as future work: (i) *grade 4*: the Wikipedia article from which the query is sampled, is regarded as the most relevant document. (ii) *grade 3*: We use the *See Also* section of a Wikipedia article in which hyperlinks link to other articles with similar or comparable information, which is mainly written manually. If there exists no *See Also* section, we use a similar section, i.e., the *Reference* section. (iii) *grade 2* and *grade 1*: Besides the *See Also* section, some hyperlinks link to pages that describe the concept of some entities in detail. We randomly sample several anchor texts from the first section and other sections, respectively, and regard the linked target pages as grade 2 and grade 1 relevant documents, respectively.

In this way, a total of 1,180,131 query-docid pairs are obtained, and we pre-train an encoder-decoder architecture using $\mathcal{L}_{total}$ as defined in Eq. (10). The architecture can be fined-tuned for downstream retrieval tasks using $\mathcal{L}_{total}$, where docids are obtained via the fusion method. We denote this version as $GR^{2P}$. In the future, we could explore using large language models to automatically label data.

## 5 Experiments

### 5.1 Experimental settings

**Datasets and evaluation metrics.** We select three widely-used multi-graded relevance datasets: Gov2 [18], ClueWeb09-B [19] and Robust04 [82]. And we use the classic normalized discounted cumulative gain (nDCG@$\{5, 20\}$), expected reciprocal rank (ERR@20) and precision (P@20) as metrics [12, 31, 49]. Furthermore, we consider two binary relevance datasets: MS MARCO Document Ranking [57] and Natural Questions (NQ 320K) [38]. We take mean reciprocal rank (MRR@$\{3, 20\}$) and hit ratio (Hits@$\{1, 10\}$) as metrics following [7, 79, 85, 104]. Following existing

Table 1: Experimental results on datasets with multi-graded relevance. Results denoted with ⋆ are from [34, 55]. And ∗,†, ‡ and ℓ indicate statistically significant improvements over the best performing SR baseline QLM, the DR baseline PseudoQ, the GR baseline RIPOR, and all the baselines, respectively ($p \leq 0.05$).

| Methods | Gov 500K | | | | ClueWeb 500K | | | | Robust04 | | | |
|---|---|---|---|---|---|---|---|---|---|---|---|---|
| | nDCG | | P | ERR | nDCG | | P | ERR | nDCG | | P | ERR |
| | @5 | @20 | @20 | @20 | @5 | @20 | @20 | @20 | @5 | @20 | @20 | @20 |
| BM25 | 0.4984 | 0.4819 | 0.5374 | 0.1848 | 0.2579 | 0.2417 | 0.3471 | 0.1362 | - | 0.4193⋆ | **0.3657**⋆ | 0.1140⋆ |
| DocT5query | 0.3936 | 0.3861 | 0.4177 | 0.1258 | 0.2071 | 0.1631 | 0.2604 | 0.0821 | 0.3613 | 0.3229 | 0.3023 | 0.1075 |
| QLM | 0.4987 | 0.4822 | 0.5379 | 0.1851 | 0.2582 | 0.2423 | 0.3475 | 0.1365 | 0.4121 | 0.4195 | 0.3658 | 0.1143 |
| SPLADE | 0.4370 | 0.4146 | 0.4445 | 0.1575 | 0.2272 | 0.2155 | 0.3050 | 0.1109 | 0.4031 | 0.3640 | 0.3192 | 0.1088 |
| RepBERT | 0.3101 | 0.3351 | 0.4305 | 0.1446 | 0.2624 | 0.2431 | 0.3650 | 0.1663 | 0.2725 | 0.2212 | 0.1686 | 0.0812 |
| DPR | 0.3236 | 0.3408 | 0.4417 | 0.1597 | 0.2614 | 0.2576 | 0.3754 | 0.1737 | 0.2873 | 0.2316 | 0.1788 | 0.0873 |
| PseudoQ | 0.4168 | 0.4383 | 0.5134 | 0.1801 | 0.2752 | 0.2704 | 0.3926 | 0.1815 | 0.4072 | 0.3577 | 0.2823 | 0.0927 |
| ANCE | 0.4152 | 0.4379 | 0.5129 | 0.1794 | 0.2743 | 0.2696 | **0.3919** | 0.1809 | 0.4069 | 0.3573 | 0.2820 | 0.0921 |
| DSI-Num | 0.2484 | 0.2647 | 0.3237 | 0.1052 | 0.1942 | 0.1690 | 0.2520 | 0.1063 | 0.2699 | 0.2028 | 0.1524 | 0.0711 |
| DSI-Sem | 0.2497 | 0.2745 | 0.3392 | 0.1215 | 0.2004 | 0.1977 | 0.2669 | 0.1143 | 0.2711 | 0.2135 | 0.1649 | 0.0737 |
| SEAL | 0.3914 | 0.3255 | 0.4418 | 0.1592 | 0.2683 | 0.2293 | 0.2927 | 0.1305 | 0.2823 | 0.2287 | 0.1654 | 0.0855 |
| DSI-QG | 0.4566 | 0.4365 | 0.4602 | 0.1702 | 0.2722 | 0.2556 | 0.3625 | 0.1783 | 0.4089 | 0.3703 | 0.3267 | 0.1032 |
| NCI | 0.4635 | 0.4473 | 0.4722 | 0.1882 | 0.2783 | 0.2631 | 0.3734 | 0.1896 | 0.4096 | 0.3786 | 0.3349 | 0.1052 |
| Ultron-PQ | 0.4658 | 0.4496 | 0.4775 | 0.1911 | 0.2798 | 0.2652 | 0.3758 | 0.1904 | 0.4103 | 0.3797 | 0.3352 | 0.1063 |
| LTRGR | 0.4663 | 0.4517 | 0.4783 | 0.1923 | 0.2805 | 0.2664 | 0.3762 | 0.1916 | 0.4109 | 0.3805 | 0.3358 | 0.1071 |
| GenRRL | 0.4669 | 0.4524 | 0.4789 | 0.1928 | 0.2812 | 0.2669 | 0.3768 | 0.1921 | 0.4112 | 0.3810 | 0.3362 | 0.1078 |
| GenRet | 0.4672 | 0.4528 | 0.4792 | 0.1931 | 0.2824 | 0.2671 | 0.3770 | 0.1925 | 0.4116 | 0.3812 | 0.3365 | 0.1081 |
| NOVO | 0.4675 | 0.4531 | 0.4796 | 0.1935 | 0.2827 | 0.2674 | 0.3372 | 0.1928 | 0.4119 | 0.3816 | 0.3369 | 0.1084 |
| RIPOR | 0.4713 | 0.4578 | 0.4831 | 0.1978 | 0.2835 | 0.2707 | 0.3401 | 0.1963 | 0.4142 | 0.3849 | 0.3404 | 0.1093 |
| $GR^{2S}$ | 0.4869ℓ | 0.4784ℓ | 0.5364ℓ | 0.2125ℓ | 0.2886∗† | 0.2791ℓ | 0.3788∗‡ | 0.2016∗† | 0.4197∗† | 0.3983ℓ | 0.3471ℓ | 0.1097† |
| $GR^{2P}$ | **0.5095**ℓ | **0.4912**ℓ | **0.5506**ℓ | **0.2167**ℓ | **0.3034**ℓ | **0.2969**ℓ | 0.3871∗‡ | **0.2026**ℓ | **0.4301**ℓ | **0.4205**ℓ | 0.3568ℓ | **0.1196**ℓ |

works [17, 74, 79, 86], for Gov2, ClueWeb09-B and MS MARCO, we primarily sampled subset datasets consisting of 500K documents for experiments, denoted as Gov 500K, ClueWeb 500K and MS 500K, respectively. For a detailed description of the datasets, please refer to Appendix D.

**Baselines.** We consider three types of baselines: sparse retrieval (SR), dense retrieval (DR), and GR models. The SR baselines include: BM25 [70], DocT5Query [28], Query Likelihood Model (QLM) [105], and SPLADE [24, 25]. The DR baselines include: RepBERT [95], DPR [36], PseudoQ [75], and ANCE [89]. The GR baselines are DSI-Num [79], DSI-Sem [79], DSI-QG [106], NCI [85], SEAL [7], GENRE [22], Ultron-PQ [104], LTRGR [44], GenRRL [102], GenRet [74], NOVO [86], and RIPOR [92]. Additionally, we compare our method with a full-ranking method, monoBERT [61]. For a detailed description of the baselines, please refer to Appendix E.

**Model variants.** We consider two versions of $GR^2$: $GR^{2S}$ and $GR^{2P}$, for supervised learning and pre-training, respectively. Additional variants are: (i) $GR^{2S}_{-RF}$ and $GR^{2P}_{-RF}$ omit the regularized fusion approach, and directly use pseudo-queries generated by a single QG model as docids. (ii) $GR^{2S}_{-\lambda}$ and $GR^{2P}_{-\lambda}$ omit the grade penalty $\lambda_l$ in $\mathcal{L}_{MGCC}$ (Eq. (9)); (iii) $GR^{2S}_{-Max}$ and $GR^{2P}_{-Max}$ omit $\mathcal{L}_{Max}$ in the MGCC loss; (iv) $GR^{2S}_{MLE}$ and $GR^{2P}_{MLE}$ only use $\mathcal{L}^q_{MLE}$ (Eq. (11)) and $\mathcal{L}^d_{MLE}$; (v) $GR^{2S}_{CE}$ and $GR^{2P}_{CE}$ use the weighted cross-entropy loss, where the relevance grades are the weights; it can be viewed as an adaption of the loss from [8, 9]; (vi) $GR^{2S}_{LR}$ and $GR^{2P}_{LR}$ directly use the LambdaRank loss [10].

**Implementation details.** For backbones, we choose the widely-used backbone in GR research, i.e., T5-base model [66] to implement the $GR^2$ and GR baselines. For docid generation, we use the docT5query model [59] as the QG model and a transformer autoencoder [80]. For T5-base, the hidden size is 768, the feed-forward layer size is 12, the number of self-attention heads is 12, and the number of transformer layers is 12. $GR^2$ and the reproduced baselines are implemented with PyTorch 1.9.0 and HuggingFace transformers 4.16.2; we re-implement DSI-Num and DSI-Sem, and utilize open-sourced code for other baselines.

For hyperparameters, we use the Adam optimizer with a linear warm-up over the first 10% steps. The learning rate is 5e-5, label smoothing is 0.1, weight decay is 0.01, sequence length of documents is

Table 2: Experimental results on datasets with binary relevance. And $*$, $\dagger$, $\ddagger$ and $\wr$ indicate statistically significant improvements over the best performing SR baseline DocT5query or SPLADE, the DR baseline PseudoQ, the GR baseline RIPOR and all the baselines, respectively ($p \leq 0.05$).

| Methods | MS 500K | | | | NQ 320K | | | |
| | MRR | | Hits | | MRR | | Hits | |
| | @3 | @20 | @1 | @10 | @3 | @20 | @1 | @10 |
|---|---|---|---|---|---|---|---|---|
| BM25 | 0.2171 | 0.2532 | 0.2385 | 0.3969 | 0.1456 | 0.1875 | 0.2927 | 0.6016 |
| DocT5query | 0.3378 | 0.3561 | 0.3489 | 0.5773 | 0.2612 | 0.2859 | 0.3913 | 0.697 |
| QLM | 0.2746 | 0.2805 | 0.2852 | 0.4593 | 0.2625 | 0.2864 | 0.3927 | 0.6979 |
| SPLADE | 0.3246 | 0.3483 | 0.3353 | 0.5637 | 0.3057 | 0.3404 | 0.4253 | 0.7146 |
| RepBERT | 0.3029 | 0.3382 | 0.3287 | 0.5233 | 0.3135 | 0.3421 | 0.4542 | 0.7275 |
| DPR | 0.3095 | 0.3264 | 0.3215 | 0.5432 | 0.3172 | 0.3493 | 0.5020 | 0.7812 |
| PseudoQ | 0.3342 | 0.3528 | 0.3452 | 0.5736 | 0.3253 | 0.3582 | 0.5271 | 0.7952 |
| ANCE | 0.3330 | 0.3520 | 0.3446 | 0.5729 | 0.3215 | 0.3576 | 0.5263 | 0.7931 |
| DSI-Num | 0.2159 | 0.2798 | 0.2676 | 0.4440 | 0.2286 | 0.2793 | 0.2185 | 0.4571 |
| DSI-Sem | 0.2229 | 0.2847 | 0.2753 | 0.4832 | 0.2581 | 0.3084 | 0.2740 | 0.5660 |
| GENRE | - | - | - | - | 0.3268 | 0.3467 | 0.2630 | 0.7120 |
| SEAL | 0.2977 | 0.3110 | 0.3072 | 0.5163 | 0.3367 | 0.3658 | 0.2630 | 0.7450 |
| DSI-QG | 0.3271 | 0.3457 | 0.3352 | 0.5749 | 0.3613 | 0.3868 | 0.6349 | 0.8236 |
| NCI | 0.3317 | 0.3566 | 0.3365 | 0.5833 | 0.3657 | 0.4053 | 0.6424 | 0.8311 |
| Ultron-PQ | 0.3326 | 0.3575 | 0.3379 | 0.5851 | 0.3663 | 0.4059 | 0.6461 | 0.8345 |
| LTRGR | 0.3354 | 0.3583 | 0.3381 | 0.5859 | 0.3692 | 0.4078 | 0.6511 | 0.8489 |
| GenRRL | 0.3359 | 0.3587 | 0.3389 | 0.5863 | 0.3698 | 0.4086 | 0.6528 | 0.8533 |
| GenRet | 0.3362 | 0.3591 | 0.3393 | 0.5867 | 0.3702 | 0.4095 | 0.6542 | 0.8567 |
| NOVO | 0.3371 | 0.3602 | 0.3405 | 0.5869 | 0.3724 | 0.4136 | 0.6613 | 0.8624 |
| RIPOR | 0.3384 | 0.3626 | 0.3421 | 0.5873 | 0.3741 | 0.4173 | 0.6638 | 0.8667 |
| $GR^{2S}$ | $0.3489^{\wr}$ | $0.3714^{\wr}$ | $0.3515^{\dagger\ddagger}$ | $0.6126^{\wr}$ | $0.3813^{\wr}$ | $0.4299^{\wr}$ | $0.6724^{\wr}$ | $0.8713^{*\dagger}$ |
| $GR^{2P}$ | $\mathbf{0.3597}^{\wr}$ | $\mathbf{0.3835}^{\wr}$ | $\mathbf{0.3821}^{\wr}$ | $\mathbf{0.6405}^{\wr}$ | $\mathbf{0.3937}^{\wr}$ | $\mathbf{0.4418}^{\wr}$ | $\mathbf{0.6832}^{\wr}$ | $\mathbf{0.8825}^{\wr}$ |

512, max training steps are 50K, and batch size is 60. We train $GR^2$ on eight NVIDIA Tesla A100 80GB GPUs. For more details, please see Appendix F.

## 5.2 Experimental results

Note, *Appendix G contains additional experimental analyses*, i.e., comparisons with full-ranking baselines (Appendix G.1) and results on large-scale datasets (Appendix G.3).

### 5.2.1 Comparison against baselines

**Results on multi-graded relevance.** Table 1 shows the performance of $GR^2$ and baselines on multi-graded relevance datasets. We find that: (i) QLM performs the best among sparse retrieval and dense retrieval baselines on Robust04 and Gov 500K, confirming prior work [45, 48, 94]; these multi-graded datasets have limited labeled training pairs, which may not be sufficient for learning semantic relationships between queries and documents. (ii) Existing GR baselines perform worse than QLM on Gov 500K and Robust04, indicating that developing an effective GR method remains an open challenge. (iii) RIPOR outperforms other GR baselines; the multi-stage training strategy appears to aid the effectiveness. (iv) By capturing the relationship between multi-graded relevance labels, $GR^2$ achieves significant improvements over GR baselines that only consider binary relevance based on the standard Seq2Seq objective. For example, $GR^{2P}$ and $GR^{2S}$ outperform RIPOR by about 11% and 14% on the Gov 500K dataset in terms of P@20, respectively. (v) Between our two methods, $GR^{2P}$ outperforms $GR^{2S}$, indicating that pre-training on large-scale elaborately constructed multi-graded relevance data, is better than training a single generation model from scratch.

**Results on binary relevance.** The performance on binary relevance datasets is shown in Table 2. We find that the relative order of different models on these datasets is almost consistent with that on the multi-graded relevance data. (i) PseudoQ outperforms QLM on these datasets. The number of labeled query-document pairs appears to be sufficient, contributing to the learning of semantic relationships. (ii) DocT5query outperforms some dense retrieval baselines on MS 500K, which may differ from

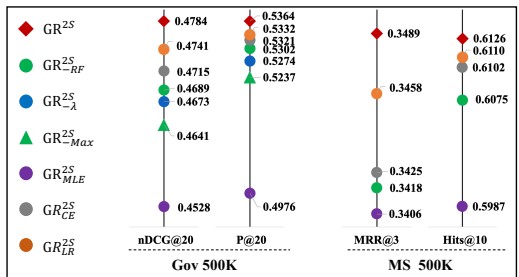 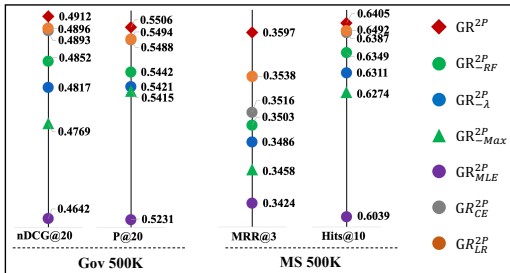

Figure 2: Ablation analysis. (Left) Supervised learning; (Right) Pre-training and fine-tuning.

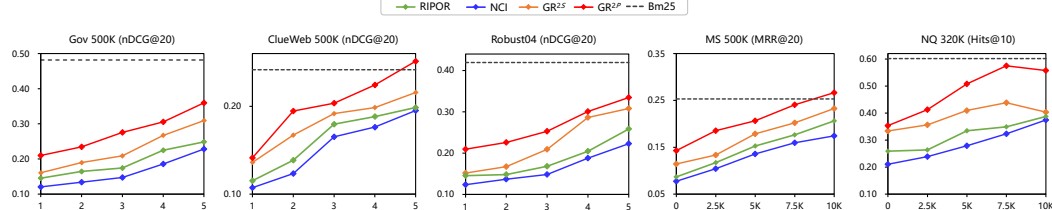

Figure 3: Supervised training and fine-tuning with limited supervision data. The x-axis indicates the number of training queries.

their performance on the full MS MARCO dataset. Similar experimental findings can be found in [74]. (iii) $GR^{2P}$ and $GR^{2S}$ perform the best, the former outperforms RIPOR by 11.7% in terms of Hits@1 on the MS 500K dataset, while the latter surpasses RIPOR by 4.3% in terms of Hits@10. This indicates that $GR^2$ is a general framework for generative retrieval that can adapt to both binary relevance and multi-graded relevance scenarios. Note, Appendix G.1 contains a comparison between $GR^2$ and the full-ranking method.

### 5.2.2 Model ablation

In Figure 2, we visualize the outcomes of our ablation analysis of $GR^2$ on Gov 500K and MS 500K.

**Docid design: regularized fusion approach.** On both datasets, (i) $GR^{2S}_{MLE}$ and $GR^{2P}_{MLE}$ outperform NCI in retrieval performance, confirming that using docids trained with the regularized fusion method is more beneficial for retrieval performance. (ii) The performance of $GR^{2S}_{RF}$ and $GR^{2P}_{RF}$ is much lower than that of $GR^{2S}$ and $GR^{2P}$, suggesting that in complex relevance scenarios, docids need to possess both relevance to the original document and distinctness.

**Training: MGCC loss.** For Gov 500K: (i) Without the grade penalty in the MGCC loss ($GR^{2S}_{-\lambda}$ and $GR^{2P}_{-\lambda}$), the query-docid pairs at different relevance grades share the same weights, which cannot fully use information about the relationships between labels. (ii) Without the grade constraint ($GR^{2S}_{-Max}$ and $GR^{2P}_{-Max}$), documents with higher relevance grades play a smaller role in optimization, weakening the discriminative ability to distinguish between grades. For MS 500K, i.e., in binary relevance scenarios, the MGCC loss in $GR^{2S}_{-\lambda}$, $GR^{2S}_{-Max}$ and $GR^{2S}$ is the supervised contrastive loss [16], showing the same performance.

For both datasets, $GR^{2S}_{CE}$ and $GR^{2P}_{CE}$ underperform $GR^{2S}$ and $GR^{2P}$, respectively. $GR^{2S}_{LR}$ and $GR^{2P}_{LR}$ show similar results. Both variants can be viewed as the pairwise approaches. They only distinguish docids at different grades, while the MGCC loss not only penalizes docids at different grades to be distinguished from each other, but also encourages docids with the same grade to be similar.

### 5.2.3 Zero-resource and low-resource settings

To simulate the low-resource retrieval setting, we randomly sample different fixed limited numbers of queries from the training set. To compare $GR^2$, NCI and RIPOR, we randomly sample 15, 30, 45 and 60 queries from multi-graded relevance datasets. For binary relevance datasets, we randomly sample 2K, 4K, 6K and 8K queries. Zero-resource retrieval is performed by only indexing without

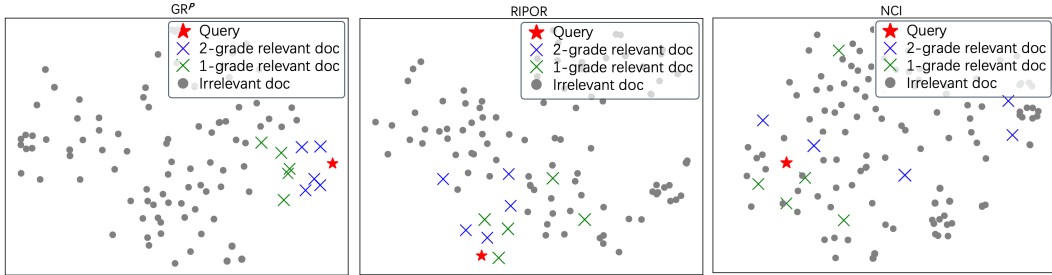

Figure 4: t-SNE plots of query and document representations for $GR^{2P}$ (left), RIPOR (mid) and NCI (right).

retrieval task, i.e., the ground-truth query-document pairs are not provided. See Figure 3. We observe the following: (i) $GR^{2S}$ and $GR^{2P}$ perform better than NCI and RIPOR, indicating that $GR^2$ is able to use relevance signals from limited information. (ii) Under the zero-resource setting, all GR methods perform worse than BM25, due to the requirements of learning the mapping between queries and relevant docids. (iii) Under the low-resource setting, on ClueWeb 500K, $GR^{2P}$ can outperform BM25 in terms of nDCG@20. $GR^{2P}$ has the pre-training stage, which helps the model to acquire a discriminative ability for relevance. In general, GR leaves considerable room for improvement under such settings; pre-training $GR^2$ with diverse corpora is likely to improve its generalization ability.

### 5.2.4  Visual analysis

We visualize query and text representations using t-SNE [81] to better understand the MGCC loss. Specifically, we sample the query "Radio station call letters" (QID: 848) from Gov 500K. We then plot a t-SNE example using the representations of the sampled query and its top-100 candidate documents given by the encoder output of $GR^{2P}$ and the representative GR baselines NCI and RIPOR. As shown in Figure 4, for $GR^{2P}$, documents with higher relevance grades are closer to the query than those with lower grades. And documents at the same relevance grade gather together. For NCI and RIPOR, the distribution of relevant documents in the latent space is relatively random: the standard Seq2Seq objective only learns to generate a single most relevant docid, from which it is difficult to learn the discriminative ability of multi-graded relevance.

### 5.2.5  Efficiency analysis

We compare the efficiency of $GR^2$ and the dense retrieval model ANCE on the Gov 500K dataset. The memory footprint refers to the amount of disk space required for storage. Additionally, we assess the end-to-end inference time during the retrieval phase. (i) Regarding memory usage, $GR^2$ primarily consists of model parameters and a prefix tree for docids. ANCE necessitates dense representations for the entire corpus, with the memory requirement increasing as the corpus size grows. Notably, $GR^2$ consumes approximately 16.7 times less memory compared to ANCE. This distinction becomes even more significant when dealing with larger datasets. For instance, in comparison to DPR, a GR approach uses 34 times less memory on the entire Wikipedia [15, 22]. (ii) In terms of inference times, the heavy process on dense vectors in dense retrieval is replaced by a lightweight generative process in $GR^2$. Consequently, $GR^2$ consumes roughly 1.59 times less inference time than ANCE. Similar efficiency gains are observed in other GR-related work [15, 74, 76].

## 6  Conclusion

We have proposed a MGCC loss for multi-graded GR that captures the relationships between multi-graded documents in a ranking, and a regularized fusion method to generate distinct and relevant docids. They work together to ensure more accurate GR retrieval. Empirical results on binary and multi-graded relevance datasets have demonstrated the effectiveness of the proposed method. There are several directions that we wish to explore: (i) We adopt hard weights for each relevance grade; what is the effect of a soft assignment setting in the MGCC loss? (ii) The generated docids remain fixed after initialization; how to perform joint optimization of the docid generation and the retrieval task? Current GR research focuses on technological feasibility, but using large language models for IR has implications for transparency, provenance, and user interactions [73]. Investigating the impact of scaled GR technology on users and societies is crucial.

## Acknowledgements

This work was funded by the National Natural Science Foundation of China (NSFC) under Grants No. 62472408 and 62372431, the Strategic Priority Research Program of the CAS under Grants No. XDB0680102, XDB0680301, the National Key Research and Development Program of China under Grants No. 2023YFA1011602 and 2021QY1701, the Youth Innovation Promotion Association CAS under Grants No. 2021100, the Lenovo-CAS Joint Lab Youth Scientist Project, and the project under Grants No. JCKY2022130C039. This research also was (partially) funded by the Hybrid Intelligence Center, a 10-year program funded by the Dutch Ministry of Education, Culture and Science through the Netherlands Organisation for Scientific Research, `https://hybrid-intelligence-centre.nl`, project nr. 024.004.022, project LESSEN with project number NWA.1389.20.183 of the research program NWA ORC 2020/21, which is (partly) financed by the Dutch Research Council (NWO), project ROBUST with project number KICH3.LTP.20.006, which is (partly) financed by the Dutch Research Council (NWO), DPG Media, RTL, and the Dutch Ministry of Economic Affairs and Climate Policy (EZK) under the program LTP KIC 2020-2023, and the FINDHR (Fairness and Intersectional Non-Discrimination in Human Recommendation) project that received funding from the European Union's Horizon Europe research and innovation program under grant agreement No 101070212, All content represents the opinion of the authors, which is not necessarily shared or endorsed by their respective employers and/or sponsors.

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

## A   Supplemental figure of the docid design

As shown in Figure 5, the regularized fusion approach includes a query generation model, i.e., docid generation model, and an autoencoder model with shared decoders, to generate relevant and distinct docids.

## B   Additional discussion

The key difference between the two regularization terms, i.e., the relevance and distinctness regularization terms, lies in the following: (i) The relevance regularization term, from a global perspective, encourages relevance between a document and its own docid, sampling docids from other documents as negative examples to ensure their non-relevance. (ii) The distinctness regularization term focuses on making documents distinguishable in the QG space (the first term of Eq. (2)) and, at the same time, ensuring distinguishability between docids in the AE space (the second term of Eq. (2)). To bridge the two spaces, we maintain a correlation between corresponding documents and docids (the third term of Eq. (2)). Although these two terms are not jointly optimized with the GR model, they can help generate more relevant and diverse docids. These fixed docids can guide the subsequent GR model towards appropriate optimization during learning, whereas joint optimization increases the learning difficulty.

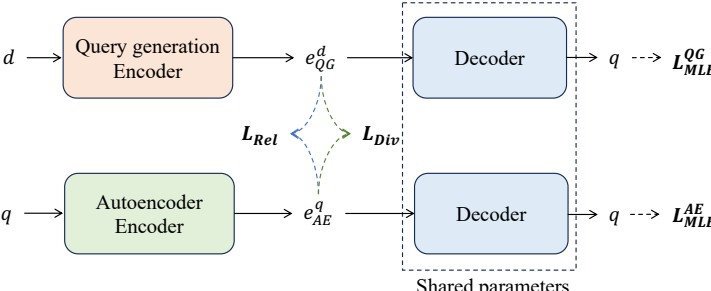

Figure 5: The regularized fusion approach to generate relevant and distinct docids.

## C   Additional related work

**Multi-graded relevance datasets.** In IR, datasets featuring multi-graded relevance annotations contribute significantly to the evaluation and advancement of retrieval models. Datasets like Robust04 [82] and large-scale web document collections such as ClueWeb09 and ClueWeb12 [19] offer diverse relevance grades. Gov2 [18] is a TREC test collection, containing a large proportion of pages in the .GOV domain. Furthermore, certain NTCIR [71] evaluation tasks contribute to the landscape.

**Sparse retrieval.** Sparse retrieval methods typically use sparse vectors to represent queries and documents and rely on exact matching to compute similarity scores, e.g., BM25 [70] and query likelihood model [39]. However, these approaches only consider statistical information and fail to incorporate semantic information. To address such limitation, some studies [3, 4, 5, 21, 26, 101] have used word embeddings to re-weight term importance. While sparse retrieval methods can reduce the computational complexity of the retrieval process, making retrieval faster and more efficient, they may fail to match at a higher semantic level, resulting in the loss of information.

**Dense retrieval.** To address the vocabulary mismatch problem [27, 99] in sparse retrieval, many researchers have turned to dense retrieval methods[47, 95]. These methods usually use a dual-encoder architecture to learn dense representations of both queries and documents, which are then processed through a matching layer to produce the final relevance score. Formally, given a query $q \in Q$ and a document $d \in D$, the two-tower retrieval model consists of two encoder functions, $\phi : Q \to \mathbb{R}^{k_1}$ and $\psi : D \to \mathbb{R}^{k_2}$, which map a sequence of tokens in $Q$ and $D$ to their corresponding embeddings $\phi(q)$ and $\psi(d)$, respectively. The scoring function $f : \mathbb{R}^{k_1} \times \mathbb{R}^{k_2} \to \mathbb{R}$ is then defined to calculate the matching score between the query embedding and the document embedding: $score(q, d) = f(\phi(q), \psi(d))$. To improve efficiency, approximate nearest neighbor (ANN) search

[2, 6] is used to accelerate retrieval. Additionally, many pre-trained models and techniques are utilized to further enhance dense retrieval performance[1, 11, 32, 37, 41, 58].

Although this pipeline paradigm has achieved promising performance, there are still some limitations: (i) Without fine-grained rerankers, such as [11, 49, 50], the performance of the vanilla dense retrieval method is still far from industrial applications. Using multiple heterogeneous modules, each with a different optimization objective, leads to sub-optimal performance. (ii) During inference, the query needs to search for relevant documents from the entire corpus. Although there are strategies for improving efficiency now available, such as ANN, such methods loss some semantic information.

**End-to-end retrieval**. Thanks to the significant success of large-scale generative models in various natural language processing tasks [42, 62, 64, 65], generative retrieval [51] has been proposed as an alternative paradigm in IR. The traditional external index is converted into a training process which learns the mapping from the documents to its docids. Given a query, a single model directly generates a list of relevant document identifiers with its parameters. This generative paradigm has potential advantages: (i) It enables end-to-end optimization towards the global retrieval objective. (ii) During inference, given a query, the model generates docids based on a small-sized vocabulary, achieving higher retrieval efficiency and eliminating the need for a heavy traditional index.

Following this blueprint, GENRE[22] was the first attempt to explore this paradigm. Using the unique titles of Wikipedia articles as document identifiers, GENRE directly generated a list of relevant article titles for a query with constrained beam search based on BART[42] model. This method outperformed some traditional pipelined methods on multiple tasks based on Wikipedia. And subsequent work [7, 15, 79, 85, 106] has continued to explore and improve upon it. However, existing work focus on binary relevance scenarios, ignoring the multi-graded relevance scenarios. In this work, we aim to further explore how generative retrieval paradigm support fine-grained relevance.

Exploring the scalability of large data and model size in GR is an area with limited research. Specifically, (i) there is work on scaling up GR to corpora in the millions [63, 92, 93], finding that that as the corpus size increases, learning difficulty rises [63]. (ii) increased model parameters enhances retrieval performance [79], which also increases latency and decreases throughput [85]. However, current research is mainly confined to datasets ranging from hundreds of thousands to millions in scale, and exploration on an even larger scale has not been undertaken. Generalizing to ultra-large-scale data is a future direction worth exploring in the field of GR.

## D   Dataset details

Multi-graded relevance datasets we used are (i) **Gov2** [18] contains about 150 queries and 25M documents collected from .gov domain web pages, from TREC Terabyte Tracks 2004–2006. Since the whole corpus is large, following existing works [17, 74, 79, 86], we primarily sampled a subset dataset consisting of 500K documents for experiments, denoted as **Gov 500K**. (ii) **ClueWeb09-B** [19] is a web collection with over 50M documents, accumulated from the TREC Web Tracks 2009–2011; we also sample a subset of 500K documents denoted as **ClueWeb 500K** to conduct experiments. (iii) **Robust04** [82] consists of 250 queries and 0.5M news articles, from the TREC 2004 Robust Track.

Furthermore, We consider two moderate-scale binary relevance datasets widely used in GR [74, 79, 85, 86, 104]: (i) **MS MARCO Document Ranking** [57] is a large-scale benchmark dataset for web document retrieval, with about 0.37M training queries; following [104], we sample a subset denoted as **MS 500K** for experiments. (ii) **Natural Questions** (NQ 320K) contains 307K query-document pairs based on the Natural Questions (NQ) dataset [38], where the queries are natural language questions and documents are gathered from Wikipedia pages. We follow the settings of existing GR work [7, 79, 85]. Table 3 shows the statistics of these datasets.

## E   Baseline details

In this section, we introduce the baselines in detail. The sparse retrieval baselines include: (i) **BM25** [70] is a classical probabilistic retrieval model. (ii) **DocT5Query** [60] generates queries conditioned on a document using T5 [65]. And these synthetic queries are then appended to the original documents. Additionally, we consider two learned sparse retrieval baselines: (iii) the Query Likelihood Model

Table 3: Data statistics. #Queries, #Documents and #Grades denote the number of labeled queries, documents and labeled relevance grades, respectively. #Avg denotes the average number of relevant documents for queries.

| Dataset | #Queries | #Documents | #Grades | #Avg |
|---|---|---|---|---|
| Gov 500K | 150 | 500K | 3 | 108 |
| ClueWeb 500K | 150 | 500K | 2 | 84 |
| Robust04 | 250 | 500K | 2 | 69 |
| MS 500K | 0.37M | 500K | 1 | 1 |
| NQ 320K | 290K | 228k | 1 | 1 |

(**QLM**) [105] adopts the query log likelihood conditioned on a document for a query as the relevance score, based on T5. (iv) **SPLADE** [24, 25] uses a BERT-based encoder to transform a text sequence into a sparse lexical representation.

The dense retrieval baselines include: (i) **RepBERT** [95] is a BERT-based two-tower model trained with in-batch negative sampling. (ii) **DPR** [36] uses dense embeddings for text segments with a BERT-based dual encoder. PseudoQ [75] generates pseudo-queries via K-means clustering based on the token embeddings of documents to enhance learning. It is also a BERT-based two-tower model. (iii) **ANCE** [89] employs an asynchronously updated approximate nearest neighbors (ANN) indexer for extracting hard negative examples to train a RoBERTa-based dual-encoder model.

The GR baselines, covering the majority of representative GR related work: (i) **DSI-Num** [79] uses arbitrary unique numbers as docids, and a MLE loss based on query-docid pairs $\mathcal{L}_{MLE}^q$ and document-docid pairs $\mathcal{L}_{MLE}^d$. (ii) **DSI-Sem** [79] builds docids by concatenating class numbers generated by hierarchical k-means clustering algorithm and adopts the same training objective as DSI-Num. (iii) **DSI-QG** [106] uses a query generation model [60] to augment the dataset, and arbitrary unique numbers as docids. (iv) **NCI** [85] uses semantic structured numbers like DSI-Sem as docids and pairs of pseudo-query and docid generated by query generation strategies for data augmentation. (v) **SEAL** [7] uses arbitrary n-grams in documents as docids and retrieves documents based on an FM-index. (vi) **GENRE** [22] retrieves a Wikipedia article by generating its title, and can only be directly used for NQ. (vii) Ultron [104] starts with pre-training using document piece-docid pairs, followed by supervised fine-tuning with labeled queries and pseudo-queries. We uses the product quantization code as docids. (viii) GenRRL [103] uses multiple optimization strategies, i.e., pointwise, pairwise, and listwise relevance signals to train the model, via reinforcement learning. We use document summaries as the docid. (ix) LTRGR [44] uses a pairwise relevance loss, e.g., margin-based rank loss to train the model. (x) GenRet [74] introduces an autoencoder model to generate discrete numbers as docids. This model is learned jointly with the retrieval task. (xi) NOVO [86] selects important word sets from the document as docids via labeled relevance signals. This method also uses pseudo-queries to augment the effectiveness.

Additionally, we compare current GR methods with full-ranking methods, since these approach typically yields better performance and are widely used. They consist of an initial retriever and a finer-grained re-ranking module. Specifically, we use a cross-encoder baseline, monoBERT [61]. BM25 retrieves the top 1000 candidate documents, and monoBERT ranks them by concatenating the query and document as the input.

# F  Additional implementation details

**Hyperparameters.**

We specify $\alpha$ and $\beta$ in Eq. (3) as 1 and 30, respectively. And the inference radius $|r|$ is set to 2 and 1.5 for the multi-graded and binary relevance datasets, respectively. Since documents in NQ have unique titles, we directly use their titles as docids. We set $\tau$ in $\mathcal{L}_{Pair}$ to 0.1, and $\gamma$ in $\mathcal{L}_{total}$ to 1. We define $\lambda_l = \frac{1}{l^2}$ used in $\mathcal{L}_{MGCC}$ following [98].

**Training.** Following [17, 74, 85, 106], we select the leading 3 paragraphs, leading 3 sentences, and randomly sample 3 entities from each document as queries, and associate them with the docid as

Table 4: Comparison between GR methods and the full-ranking baseline. $*$ indicates statistically significant improvements over $\mathrm{GR}^{2P}$ ($p \leq 0.05$).

| Methods | Gov 500K | MS 500K |
|---|---|---|
| | nDCG@5 | MRR@20 |
| RIPOR | 0.4713 | 0.3626 |
| $\mathrm{GR}^{2P}$ | 0.5095 | 0.3835 |
| BM25+monoBERT | **0.6953**$^*$ | **0.5862**$^*$ |

additional query-docid pairs for data augmentation. And inspired by [92], to enhance the model's effectiveness, we also adopt self-negative fine-tuning strategy.

**Inference.** During inference, we construct a prefix trie [22] for all docids, and adopt constrained beam search to decode docids with 20 beams.

**Safeguards for the responsible release of resources.** For pretrained language models, rigorous evaluation and testing protocols are employed to assess potential risks and biases before release. Additionally, we plan to outline strict guidelines for access control and usage policies to mitigate misuse upon publication. Regarding data release, anonymization techniques are used to protect privacy, and sensitive information is redacted or excluded where necessary upon publication.

## G   More experimental results

### G.1   Comparison between GR methods with the full-ranking baseline

From Table 4, we observe that GR is still in its early stages, and the current GR methods have some distance to cover compared to full-ranking methods. Achieving the integration of index, retrieval, and reranking into a single model poses a significant challenge. Possible reasons include (i) the design of docids is still independent of the final retrieval optimization; (ii) there is a lack of explicit interaction between query and document; (iii) the current optimization methods do not fully use the data; and (iv) the pre-training tasks for backbone models are not specifically designed for GR. Current GR methods can only be compared with index-retrieval frameworks, and there is still some distance from the ideal model-based IR. We hope that future work will address these aspects.

### G.2   Efficiency analysis

We compare the efficiency of $\mathrm{GR}^2$ and the dense retrieval model ANCE on the Gov 500K dataset. The memory footprint refers to the amount of disk space required for storage. Additionally, we assess the end-to-end inference time during the retrieval phase. (i) Regarding memory usage, $\mathrm{GR}^2$ primarily consists of model parameters and a prefix tree for docids. ANCE necessitates dense representations for the entire corpus, with the memory requirement increasing as the corpus size grows. Notably, $\mathrm{GR}^2$ consumes approximately 16.7 times less memory compared to ANCE. This distinction becomes even more significant when dealing with larger datasets. For instance, in comparison to DPR, a GR approach uses 34 times less memory on the entire Wikipedia [15, 22]. (ii) In terms of inference times, the heavy process on dense vectors in dense retrieval is replaced by a lightweight generative process in $\mathrm{GR}^2$. Consequently, $\mathrm{GR}^2$ consumes roughly 1.59 times less inference time than ANCE. Similar efficiency gains are observed in other GR-related work [15, 74, 76].

### G.3   Results on large-scale datasets

Generalization on large-scale datasets for GR remains a significant challenge [63, 77], which requires dedicated design and research, with only a few works exploring it [63, 92, 93]. Although they can achieve comparable effectiveness to dense retrieval methods, their performance still lags behind mainstream full-ranking methods. Therefore, current efforts are still focused on moderate-scale datasets [43, 74, 79, 86, 104]. Although generalization is not the main focus of this work, we still conduct some preliminary experiments. In the future, we plan to investigate how to generalize to extremely large-scale datasets, such as the corpora in the industry with billions of documents.

Table 5: Results on the MS 1M dataset.

| Methods | MS 1M |
| --- | --- |
| | MRR@20 |
| RIPOR | 0.3674 |
| GR$^{2P}$ | 0.3659 |

To assess the performance of our method on million-scale datasets, specifically, we constructed a corpus of size 1M based on the MS MARCO dataset, referred to as MS 1M. Note, it is a binary relevance dataset, with annotation data and validation sets directly derived from the original dataset. Initially, we pre-train GR$^{2P}$ on the constructed 4-grade relevance Wikipedia dataset and then fine-tune and evaluate on MS 1M. We compare the performance of GR$^{2P}$ with RIPOR on MS 1M.

As shown in Table 5, we observe that (i) RIPOR performs well on large-scale datasets and exhibits strong generalization, possibly attributed to its docids generated by its own GR model, along with multi-stage enhancement training. (ii) Additionally, our method achieves comparable results to RIPOR on such a large-scale corpus, indicating that our method considering multi-graded relevance information also contributes to more accurate relevance distinction on binary relevance datasets.

