# OpenReview forum: "Generative Retrieval Meets Multi-Graded Relevance"
_NeurIPS.cc/2024/Conference — NeurIPS 2024 spotlight_

### Official Review · Reviewer_KSwY · 2024-07-12

**Soundness:** 3
**Presentation:** 2
**Contribution:** 3
**Rating:** 7
**Confidence:** 3

**Summary:**

The paper targets the problem of Multi-Graded Relevance in a Generative Retrieval setup.

Generative Retrieval uses a Seq2Seq model to decode relevant “docids” for a given query. Prior work on Generative Retrieval has focused on tasks where there is only one relevant document per query and the degree of relevance is binary : “relevant” or “not-relevant”. The paper talks about Multi-graded Retrieval tasks, where each query has multiple relevant documents, each having its own degree of relevance.

The paper proposes a methodology to tackle the graded retrieval tasks in a Generative Retrieval Framework. It identifies the key challenges : 1. Having distinct but relevant “docids” and 2. Graded Relevance when matching query and documents. It proposes solutions to these challenges in “regularized fusion” and “MGCC” approaches respectively.

Additionally the authors also develop and experiment with a pre-training setup for multi-graded relevance retrieval.

**Strengths:**

1. The problem of multi-graded retrieval in Generative Retrieval is relevant/important with no prior work
2. The challenges are well-identified i.e., the goal of having distinct but relevant docids for documents.
3. The solutions proposed are well formulated and intuitive to understand.
4. Empirical Results are strong with thorough ablations and detailed discussions.
5. The authors propose and experiment with a pre-training framework, showing positive results.

**Weaknesses:**

While not a strong weakness, the mathematical notation in the paper can use some work to make the paper an easier read. I found it tough to parse through “Section 3 : Preliminaries”. The paper would benefit from resolving this.

**Questions:**

I just have a few clarifying questions :
1. Are there numbers on Recall? Prior work I believe focuses on MRR and Recall metrics.
2. Why include the last term in Eq2 (similarity in embedding space) when the same is taken care of by Eq1? Is it for training stability?
3. If I understood correctly, then docids are generated by decoding a random sample on the hypersphere  (z=e+r). Does that mean if we were to sample multiple times for each document, we would get distinct docids from the randomness in r itself? Do we need e of documents (through QG module) to be “distinct” (i.e. do we need regularized fusion if we have better sampling or more samples of r?)

**Limitations:**

Typo : In line 188, “These fixed” needs “T” to be lowercase

While not a limitation, I believe the paper would also benefit from a few Qualitative examples showing Multi-graded relevance in practice. An uninitiated reader might not be familiar with Generative Retrieval or Multi-graded relevance and a figure/qualitative example might be useful for them.

---

> ### Author Rebuttal · Authors · 2024-08-06
>
> Thank you for the constructive and valuable comments. With regard to different comments about weaknesses, our responses are as follows:
>
> **[Comment 1]** While not a strong weakness, the mathematical notation in the paper can use some work to make the paper an easier read. I found it tough to parse through “Section 3 : Preliminaries”. The paper would benefit from resolving this.
>
> **[Response 1]** Thank you for your suggestions. We will simplify the mathematical notation in the paper and add a notation table.
>
> ---
> **[Comment 2]** Are there numbers on Recall? Prior work I believe focuses on MRR and Recall metrics.
>
> **[Response 2]** Thank you for your question. Since the Hits metric and the Recall metric are very similar in principle, we did not include Recall separately. Similarly, [79, 106, 7, 17, 102] only used Hits or used both Hits and MRR.
>
> ---
> **[Comment 3]** Why include the last term in Eq2 (similarity in embedding space) when the same is taken care of by Eq1? Is it for training stability?
>
> **[Response 3]** Thank you for your question.
>
> (1) For both the relevance and distinctness regularization terms, the similarity is used to ensure relevance between the document and the query, i.e., the docid (for example, the numerator in Eq 1 and the numerator of the last term in Eq 2). Additionally, the denominator in Eq 1 ensures that irrelevant queries and documents are dissimilar.
>
> (2) The first two terms in Eq 2 push apart different document representations encoded by the QG model and different query representations encoded by the AE model. To bridge the QG and AE latent spaces, the last term in Eq 2 builds upon the first two terms by making relevant document and query representations similar. Note that it does not involve comparing documents to irrelevant queries, which is a distinction from Eq 1.
>
> ---
> **[Comment 4]** If I understood correctly, then docids are generated by decoding a random sample on the hypersphere (z=e+r). Does that mean if we were to sample multiple times for each document, we would get distinct docids from the randomness in r itself? Do we need e of documents (through QG module) to be “distinct” (i.e. do we need regularized fusion if we have better sampling or more samples of r?)
>
> **[Response 4]** Thank you for your question.
>
> (1) Sampling multiple times can indeed yield different docids. However, our requirement for docids is not just simple "distinctness," but to ensure that docids are relevant to the documents and that they are "distinct" based on semantic differences among the documents. Achieving this requires using e as an anchor, which is why regularized fusion is necessary.
>
> (2) Sampling strategies are heuristic post-processing methods and might not guarantee that the model generates high-quality docids. Regularized fusion, on the other hand, can optimize the model to ensure better generation capability.
>
> ---
> **[Comment 5]** Typo : In line 188, “These fixed” needs “T” to be lowercase
>
> **[Response 5]** Thank you for pointing this out. We will make the correction.
>
>
>
> ---
> **[Comment 6]** The paper would also benefit from a few Qualitative examples showing Multi-graded relevance in practice. An uninitiated reader might not be familiar with Generative Retrieval or Multi-graded relevance and a figure/qualitative example might be useful for them.
>
> **[Response 6]** Thank you very much for your suggestions. We add some quantified examples of multi-graded relevance as the follows.
>
> In real-world search scenarios, documents might be described with different degrees of relevance [Yu et al., 2009; Scheel et al., 2011] with respect to queries, such as not relevant, partially relevant, and highly relevant. For example, the Yahoo! search engine presented "Perfect" relevant documents at the top in the ranking, followed by "Excellent", "Good", and "Fair" relevant documents, without showing "Bad" relevant documents [Chapelleet al., 2011]. There are also some other examples include medical retrieval [Yu et al., 2009], recommender systems [Scheel et al., 2011], as well as representative retrieval benchmarks [Tetsuyaal., 2005;18,19].
>
> Targeting multi-graded instead of simple binary relevance, helps to meet the demand of practical use cases, while the collection of multi-graded annotations helps to improve the retrieval results.  Accordingly, the importance of IR evaluation based on multi-graded relevance assessments is increasingly attracting attention; see, e.g., the NII Test Collection for IR Systems (NTCIR) project [71] and many widely-used ranking metrics [12,31,49].
>
> - [Yu et al., 2009] Enabling Multi-Level Relevance Feedback on Pubmed by Integrating Rank Learning into DBMS
> - [Scheel et al., 2011] Performance Measures for Multi-Graded Relevance
> - [Chapelleet al., 2011] Yahoo! Learning to rank challenge overview
> - [Tetsuya et al., 2005] Ranking the NTCIR Systems Based on Multigrade Relevance

---

> > ### Comment · Reviewer_KSwY · 2024-08-12
> >
> > Thank the authors for the detailed response.

---

### Official Review · Reviewer_xXNY · 2024-07-13

**Soundness:** 3
**Presentation:** 3
**Contribution:** 3
**Rating:** 7
**Confidence:** 3

**Summary:**

The work deals with generative retrieval for cases where documents have multi-graded relevance. The authors propose GR2 a framework for generative retrieval in such cases by tackling two important challenges. First they optimize for relevance and distinctiveness of document IDs by a regularized fusion approach which comprises a pseudo query generation module followed by an auto-encoding module that reconstructs the document ids. Secondly, the authors introduce a graded, constrained contrastive learning approach which aims to bring together representations of queries with the representations of the relevant docids and push apart irrelevant docids in mini-batch. GR2 shows impressive performance improvements over other approaches and is also relatively efficient.

**Strengths:**

- The work tackles an important task of generative retrieval for cases of documents with multi-graded retrieval. Compared to existing works like [2], the work captures the relationship between labels using a graded contrastive loss leading to impressive gains over existing generative IR models and dense IR models.
- The loss functions are well motivated and shows the impact of contrastive learning on generative retrieval. I think further studies on effect of different contrastive losses would be an interesting avenue to explore.
- The authors perform extensive experiments and ablations of the proposed method to reinforce the importance of each of the components.


[1] Approximate Nearest Neighbor Negative Contrastive Learning for Dense Text Retrieval LEE et. al.
[2] Learning to tokenize for generative retrieval.

**Weaknesses:**

- Some important baselines are missing in the current version of the work. ColBERT (ColBERTV2) in PLAID setup is a powerful dense retriever and  would serve as a valuable approach for comparison. Additionally,  dense retrieval models like contriever and tas-b have also demonstrated impressive performance on a wide range of benchmarks and are strong dense retrievers due to their training objectives and should be included for comparison. Particularly since GR2 is also used in pre-training setup it is important to compare to retrievers like Contriever. pre-trained using contrastive learning for retrieval.


- While there is clear intuition for the distinctiveness, relevance regularization losses and the MGCC loss, further experiments could be carried out with respect to the contrastive learning parts of these approaches  to understand the approach better and  to help improve or strengthen the results. For instance, negative sampling is an important aspect of contrastive learning and a good mix of hard and soft negatives are crucial for learning useful representations. While in MGCC authors explore mini-beach based negatives, it is also important to explore global negatives similar to works like ANCE [1].

- The paper would benefit from some error analysis. This involves analysis of cases where wrong docids are considered relevant, and an analysis of whether it stems from the stochasticity of generative models or other external factors. The paper does not currently report any list of common errors or a qualitative analysis of results. It is critical to understand where the current approach fails as it would provide avenues for further research.


[1] Approximate Nearest Neighbor Negative Contrastive Learning for Dense Text Retrieval LEE et. al.
[2] Learning to tokenize for generative retrieval.

**Questions:**

- Equation 10 gives equal weights to the regularization losses. Have you considered weighting them differently ?

**Limitations:**

While authors briefly describe the limitations in conclusion a more detailed explanation would help contextualize the work and understand future directions better.

---

> ### Author Rebuttal · Authors · 2024-08-06
>
> Thank you for the constructive and valuable comments. With regard to different comments, our responses are as follows:
>
> **[Comment 1]** Some important baselines are missing in the current version of the work. ColBERT (ColBERTV2) in PLAID setup is a powerful dense retriever and would serve as a valuable approach for comparison. Additionally, dense retrieval models like contriever and tas-b have also demonstrated impressive performance on a wide range of benchmarks and are strong dense retrievers due to their training objectives and should be included for comparison.
>
> **[Response 1]** Thank you for your suggestions.
> Due to the time constraints of the rebuttal, we have now included the effects of ColBERT and Contriever on the Gov 500K dataset, as shown in the table below:
>
> | Method|nDCG@5| nDCG@20|P@20 |ERR@20|
> |---|---|---|---|---|
> | ColBERT| 0.4384|0.4529|0.5162 | 0.1803|
> | Contriever|0.4683| 0.4641| 0.5267| 0.1979|
> |GR$^{2S}$|0.4869|0.4784|0.5364|0.2125|
> |GR$^{2P}$|0.5095|0.4912|0.5506|0.2167|
>
> We can observe that these two baselines perform better than strong GR baselines, such as NOVO, but their performance is slightly worse than RIPOR and GR$^2$, which also validates the effectiveness of our method.
>
> ---
>
> **[Comment 2]** While there is clear intuition for the distinctiveness, relevance regularization losses and the MGCC loss, further experiments could be carried out with respect to the contrastive learning parts of these approaches to understand the approach better and to help improve or strengthen the results. For instance, negative sampling is an important aspect of contrastive learning and a good mix of hard and soft negatives are crucial for learning useful representations. While in MGCC authors explore mini-beach based negatives, it is also important to explore global negatives similar to works like ANCE.
>
> **[Response 2]** Thank you very much for your feedback.
>
> (1) If we remove the contrastive learning from the distinctiveness and relevance regularization losses, these two regularization terms will degrade to maximizing the similarity between the query and relevant documents. We denote this variant as GR$^{2S}_{Sim}$.
>
> If we remove the contrastive learning from the MGCC loss, this loss will also degrade to maximizing the similarity between the query and related documents, but with an additional relevance grade weight, which is exactly the GR$^{2S}_{CE}$ variant we explored in line 271. The performance of these two variants compared to GR$^2$ on Gov500K is as follows:
>
> | Method |nDCG@20| P@20 |
> |---|---|---|
> | GR$^{2S}_{Sim}$| 0.4693|0.5308|
> | GR$^{2S}_{CE}$|0.4715| 0.5321|
> |GR$^{2S}$|0.4784|0.5364|
> |GR$^{2P}$|0.4912|0.5506|
>
> We can see that both variants perform worse than GR$^{2}$, which validates the effectiveness of contrastive learning in these losses.
>
> (2) We also agree that global negatives might be more beneficial for performance. Due to hardware constraints, we have used contrastive learning within the mini-batch range. One possible approach to utilize global negatives is to first perform warm-up training with mini-batch negatives, then use the resulting model to filter hard negatives on a global range, and finally use them for a second phase of enhanced training. However, this approach might increase the optimization cost. We will explore more efficient methods in future work.
>
> ---
> **[Comment 3]** The paper would benefit from some error analysis.
>
> **[Response 3]** Thank you for your suggestions.
>
> (1) We performed an error analysis on GR$^{2P}$. Specifically, in the MS 500K dataset, we identified a bad case where the query (Query ID: 1054438) is "explain grievances". The relevant ground-truth document's docid is “what is grievance?”. In the list of docids predicted by GR$^{2P}$, the top-1 docid is an irrelevant one: “explain complaints”, with the ground-truth docid ranked second. Both documents are semantically similar.
> The reason GR$^{2P}$ failed to rank the correct docid first might be due to our sequence-based docid approach, which considers order and requires exact generation. The targeted document will be missed from the retrieval results if any step of the generation process makes a false prediction about its identifier [96]. This issue arises from using a prefix tree to constrain decoding, which is a common problem in many GR studies [79, 106, 85, 22, 104]. One possible improvement is to use term-sets for decoding constraints, but this requires more storage space [96].
>
> (2) We will add this analysis to the appendix in our subsequent revisions.
>
> ---
> **[Comment 4]** Equation 10 gives equal weights to the regularization losses. Have you considered weighting them differently ?
>
> **[Response 4]** Thank you very much for your question.
> We experimented with different values for $\gamma$, while keeping the weights of the MLE losses in Equation 10 as 1. Our experiments revealed that:
>
>  (1) When $\gamma$ is less than 1, there is a slight decrease in performance. This may be because it weakens the effect of the MGCC loss, causing the model to learn less effectively about relevance.
>
> (2) When $\gamma$ is greater than 1, the performance decreases more significantly. This is likely because it weakens the effect of the other two losses, which are fundamental operations in GR, namely indexing and retrieval. Poor learning of these operations has a significant impact on performance. Therefore, we set $\gamma$ to 1.
>
> ---
> **[Comment 5]** While authors briefly describe the limitations in conclusion a more detailed explanation would help contextualize the work and understand future directions better.
>
> **[Response 5]** Thank you for your suggestions. We will add a description of the limitations in the appendix in our subsequent revisions.

---

### Official Review · Reviewer_VoH6 · 2024-07-13

**Soundness:** 3
**Presentation:** 4
**Contribution:** 3
**Rating:** 7
**Confidence:** 4

**Summary:**

The paper proposes a novel QG based docid for generative retrieval, optimising relevance and distinctness of the generated queries jointly. It introduces the MGCC loss with multi-graded labels. Experiments on subsets of Gov2, ClueWeb09-B, Robust04, MS Marco, and NQ with up to 500k documents validate the effectiveness of the methods.

Overall, the contributed methods are valuable and inspiring, and the results are convincing.

**Strengths:**

- Jointly optimizing relevance and distinctness with an AE block is a nice idea, effectively balancing both factors as demonstrated by ablation studies.
- Creating semi-supervised graded PT data from Wikipedia seems very useful, with remarkable improvements from this additional training.
- The paper is well-presented and the experiments are comprehensive.

**Weaknesses:**

- Limited corpus scale remains a concern for various GR works, further discussion on its potential influence on the proposed method is needed. Nonetheless, a 500k corpus is substantial and applicable.
- The paper should discuss the existing works more about the use of QG in GR

**Questions:**

- The term "pre-train" is somewhat misleading, as using semi-supervised Wikipedia data for further training differs from typical LLM pre-training.
- Some examples for generated queries, both with and without using the RF building block, to better understand the reasons behind its effectiveness.

**Limitations:**

Yes

---

> ### Author Rebuttal · Authors · 2024-08-06
>
> We sincerely appreciate your constructive and valuable comments. In response to the various feedback, we address each point as follows:
>
> **[Comment 1]** Limited corpus scale remains a concern for various GR works, further discussion on its potential influence on the proposed method is needed. Nonetheless, a 500k corpus is substantial and applicable.
>
> **[Response 1]** Thank you for your suggestions. We agree that handling large-scale corpora, such as those in the millions, is indeed necessary and represents a significant challenge for GR. There is currently limited research exploring this issue, and we discuss relevant work in lines 731-737 of Appendix C. Additionally, we are actively exploring ways to better manage and remember corpora of this scale.
>
> ---
> **[Comment 2]** The paper should discuss the existing works more about the use of QG in GR
>
> **[Response 2]** Thank you very much for your suggestions.
>
> (1) Currently, GR has utilized QG in the following works: (i) DSI-QG [106], NCI [85], and RIPOR [92] use QG as a data augmentation technique to generate pseudo-queries as the input. (ii) [43, 76] use pseudo-queries as one of the identifiers. For the second type, our method differs in that they directly use pseudo-queries generated by DocT5query as identifiers without additional optimization, which results in multiple semantically similar documents sharing the same identifier.
>
> (2) We have a brief description in line 93. We will provide further details in the related work section.
>
>
> ---
>
> **[Comment 3]** The term "pre-train" is somewhat misleading, as using semi-supervised Wikipedia data for further training differs from typical LLM pre-training.
>
> **[Response 3]** Thank you for pointing this out. Indeed, our "pre-training" differs from typical LLM self-supervised pre-training. The reason we describe it this way is to follow the work of [104, 15], where they construct pseudo data pairs for training to enhance the model's capability for retrieval tasks and better align with downstream retrieval tasks.
>
> ---
> **[Comment 4]** Some examples for generated queries, both with and without using the RF building block, to better understand the reasons behind its effectiveness.
>
> **[Response 4]** Thank you very much for your suggestions.
>
> (1) We sampled a query (Query ID: 289812) from the MS 500K dataset: "How many mm is a nickel coin?". The top-3 docids generated by GR$^{2S}$ and GR$^{2S}_{RF}$ are as follows (correct predictions are indicated in italics):
>
> | Rank | GR$^{2S}$ | GR$^{2S}_{RF}$ |
> |---|---|---|
> |1| *What is the diameter of a nickel coin in millimeters?* | How much was nickel in a coin?|
> |2| How much was nickel in a coin? |How heavy is the nickel in grams?|
> |3| What was the weight of the nickel coin?| When was nickel silver first used?|
>
> We can observe that GR$^{2S}$ rank the correct docid first. However, the top 3 identifiers predicted by GR$^{2S}_{RF}$ do not include the correct docid, even though all the predicted docids share keywords with the query, such as "nickel." There is still a gap in relevance compared to the correctly predicted docid, which further validates the importance of the RF module for docid quality.
>
> (2) We will add this case study to the experimental results section in Section 5.

---

### Official Review · Reviewer_UkCg · 2024-07-22

**Soundness:** 3
**Presentation:** 3
**Contribution:** 3
**Rating:** 5
**Confidence:** 4

**Summary:**

In this paper, the authors propose a new generative retrieval model, which utilizes multi-grade relevance labels instead of binary relevance. Using graded relevance labels is not well-discussed in previous works.

Pros:
- The problem itself is interesting and important. Generative retrieval is a hot research topic in the IR community.
- The proposed solution is reasonable.


Cons:
- In the Introduction, the authors introduced the reason why the simple generation likelihood of docids cannot work for graded relevance.  It is strange to claim that "Docids commonly exhibit distinct lengths" given the truth that some existing docid schemas (such as semantic id used in DSI, the PQ-based ID, etc.) Even for token-based docID, we can still add a fix-length constraint. It is hard to convince the readers by claiming that "as a fixed length might not adequately encompass diverse document semantics".
- Both DocID schema design and learning from multi-grade relevance are considered in the paper. I am quite interested in learning whether the multi-grade relevance part can also work with other DocId schemas. So an ablation study about this (e.g., using semantic string as docids) should be given (note that this is different from the experiments in Section 5.2.2) .

**Strengths:**

- The problem itself is interesting and important. Generative retrieval is a hot research topic in the IR community.
- The proposed solution is reasonable.

**Weaknesses:**

Cons:
- In the Introduction, the authors introduced the reason why the simple generation likelihood of docids cannot work for graded relevance.  It is strange to claim that "Docids commonly exhibit distinct lengths" given the truth that some existing docid schemas (such as semantic id used in DSI, the PQ-based ID, etc.) Even for token-based docID, we can still add a fix-length constraint. It is hard to convince the readers by claiming that "as a fixed length might not adequately encompass diverse document semantics".
- Both DocID schema design and learning from multi-grade relevance are considered in the paper. I am quite interested in learning whether the multi-grade relevance part can also work with other DocId schemas. So an ablation study about this (e.g., using semantic string as docids) should be given (note that this is different from the experiments in Section 5.2.2) .

**Questions:**

na

---

> ### Author Rebuttal · Authors · 2024-08-06
>
> Thank you for the constructive and valuable comments. With regard to your comments, our responses are as follows:
>
> **[Comment 1]** In the Introduction, the authors introduced the reason why the simple generation likelihood of docids cannot work for graded relevance. It is strange to claim that "Docids commonly exhibit distinct lengths" given the truth that some existing docid schemas (such as semantic id used in DSI, the PQ-based ID, etc.) Even for token-based docID, we can still add a fix-length constraint. It is hard to convince the readers by claiming that "as a fixed length might not adequately encompass diverse document semantics".
>
> **[Response 1]** Thank you for pointing this out.
> 1. Indeed, the current design of docids can be either fixed-length or variable-length. Among variable-length designs, there are about 8 types, such as unstructured atomic integers[79], naively structured strings[79], titles[22], URLs[104], pseudo queries[76], important terms[14], n-grams[7], and multi-view identifiers[43]. Fixed-length designs are roughly 4 types: semantically structured strings[79], PQ-based strings[104,92], learnable numbers[74], and learnable n-gram sets[86]. We stated that "Docids commonly exhibit distinct lengths" because variable-length identifiers typically have a lower acquisition cost and are thus widely used. In contrast, fixed-length learnable identifiers offer better retrieval performance but usually require more complex learning tasks and the optimization process is more challenging [74, 86, 77].
>
> 2. Regarding the "token-based docID" you mentioned, it is indeed possible to directly set a fixed-length constraint. However, setting a heuristic fixed value might lead to information loss (if the length is too short) or excessive storage cost (if the length is too long). Additionally, different datasets may require different suitable values. Therefore, we did not adopt this approach.
>
> 3. To describe more accurately, we will modify the statement (Line 43) to: “For variable-length token-based docids, such as titles, URLs, etc., which are easily obtainable, it may be challenging to comprehensively include the diverse information of the document. For fixed-length learnable docids, obtaining such docids incurs higher optimization costs. Considering the the cost of genearating docids, this work focuses on variable-length token-based docids.”
>
> ---
> **[Comment 2]** I am quite interested in learning whether the multi-grade relevance part can also work with other DocId schemas. So an ablation study about this (e.g., using semantic string as docids) should be given.
>
> **[Response 2]** Thank you for your question.
> If semantic strings are used as docids, they can also be applied to our MGCC loss. This type of identifier inherently possesses a degree of relevance to the document and distinctness between docids. Therefore, we directly follow the method in DSI[79] to generate semantic strings as docids.
> We combine this docid with the MGCC loss and optimize it using supervised learning, denoted as GR$^{2S}_{sem}$. The performance on the Gov 500K dataset is as follows:
>
> | Variant | nDCG@5 | nDCG@20 | P@20 | ERR@20 |
> |----------|----------|----------|----------|----------|
> | GR$^{2S}_{sem}$ | 0.4264 | 0.3487 | 0.4618 | 0.1893|
> |GR$^{2S}$|0.4869|0.4784|0.5364|0.2125|
> |GR$^{2P}$|0.5095|0.4912|0.5506|0.2167|
>
>
>
>
> We can observe that the performance of GR$^{2S}_{sem}$ is significantly worse compared to our GR$^{2S}$. The reason might be that such docids have a greater gap compared to token-based ids and queries or documents, making the learning process for the model more challenging (which is consistent with the findings in [7, 22]). Additionally, their distinctness is determined by a combination of clustering indices and randomly assigned numbers within the final layer clusters (for algorithm details, please refer to the DSI[79] paper). In other words, these indices better reflect similarities (the same numbers have similar semantics), which to some extent weakens the differences (the degree of difference between different indices does not necessarily correlate strongly with the degree of difference in document content).

---

> > ### Comment · Reviewer_UkCg · 2024-08-10
> >
> > Thank the authors for the detailed response.

---

### Decision · Program_Chairs · 2024-09-25

**Decision:**

Accept (spotlight)

**Comment:**

The reviewers agreed that multi-graded relevance in generative retrieval systems is an important and largely unexplored research direction, that the proposed approach was novel, the proposed solution reasonable, and the experimental results strong.

To further improve the paper, authors are encouraged to consider the more critical feedback, including improving the clarity of the writing and possibly including additional experimental results, such as additional baselines, more ablations, and detailed error analysis.

Please see the individual reviews for additional feedback and suggestions.